# Assessing Factors Associated with Non-Fatal Injuries from Road Traffic Accidents among Malaysian Adults: A Cross-Sectional Analysis of the PURE Malaysia Study

**DOI:** 10.3390/ijerph19148246

**Published:** 2022-07-06

**Authors:** Zaleha Md Isa, Noor Hassim Ismail, Rosnah Ismail, Azmi Mohd Tamil, Mohd Hasni Ja’afar, Nafiza Mat Nasir, Maizatullifah Miskan, Najihah Zainol Abidin, Nurul Hafiza Ab Razak, Khairul Hazdi Yusof

**Affiliations:** 1Department of Community Health, Faculty of Medicine, UKM Medical Centre, Universiti Kebangsaan Malaysia, Cheras, Kuala Lumpur 56000, Malaysia; zms@ppukm.ukm.edu.my (Z.M.I.); drrose@ppukm.ukm.edu.my (R.I.); drtamil@ppukm.ukm.edu.my (A.M.T.); drmhasni@ukm.edu.my (M.H.J.); najihahza@gmail.com (N.Z.A.); hafiza@ukm.edu.my (N.H.A.R.); hazdi@ppukm.ukm.edu.my (K.H.Y.); 2Department of Primary Care Medicine, Faculty of Medicine, Universiti Teknologi MARA, Sungai Buloh Campus, Selangor Branch, Sungai Buloh 47000, Selangor, Malaysia; drnafiza@uitm.edu.my; 3Department of Primary Care Medicine, Faculty of Medicine and Defence Health, National Defence University of Malaysia, Sungai Besi, Kuala Lumpur 57000, Malaysia; maizatullifah@upnm.edu.my

**Keywords:** non-fatal injuries, road accident, PURE, siesta, public health

## Abstract

Non-fatal injuries (NFIs) due to road traffic accidents (RTAs) are a public health problem worldwide that significantly impacts the population morbidity and healthcare costs. As the demands for vehicles in developing countries, such as Malaysia, is increasing annually, the present study aims to determine the prevalence and factors associated with NFIs due to RTAs among Malaysia’s adult population. Methods: This was a cross-sectional study involving 15,321 participants from the Prospective Urban and Rural Epidemiological (PURE) study conducted in Malaysia. Participants reported whether they had experienced an NFI that limited their normal activities within the past 12 months. Data on risk factors for NFIs were elicited. Multiple logistic regression models were fitted to identify the associated factors. Results: Overall, 863 participants (5.6% of 15,321) reported at least 1 NFI in the past 12 months, with 303 caused by RTAs (35.1%), 270 caused by falls (31.3%) and 290 attributed to other causes (33.6%). The factors associated with higher odds of sustaining an NFI due to an RTA were being male (adjusted odd ratio (AOR) 2.08; 95% CI 1.33–3.26), having a primary (2.52; 1.40–4.55) or secondary (2.64; 1.55–4.49) level of education, being overweight to obese (1.40; 1.01–1.94), being currently employed (2.03; 1.31–3.13) and not practicing a noon nap/siesta (1.38; 1.01–1.89). Conclusions: The occurrence of NFIs due to RTAs is highly preventable with strategic planning aimed at reducing the risk of RTAs among the Malaysian population. Interventions focusing on protecting road users, especially those who drive two-wheelers, with proactive road safety awareness and literacy campaigns, combined with strict enforcement of the existing traffic laws and behavioural modifications, may reduce the risk of NFIs following RTAs.

## 1. Introduction

As a developing country, the transportation system in Malaysia has become vital, and the demand for vehicles has been increasing every year [1,2]. The number of registered vehicles was 15,790,732 in 2006, and it grew to 29,666,187 by September 2018 [3,4]. The rising population and number of vehicles on the roads increased the risk of road traffic accidents (RTAs), leading to injuries and even fatalities [5,6]. Data from the Ministry of Transport (MOT) showed a steady increase in the number of RTAs reported in Malaysia during the last 10 years, from 414,421 in 2010 to 567,516 in 2019. In 2019 alone, more than MYR 9 billion was spent on medical costs related to RTAs [7,8].

Several local studies have reported that the road accidents that led to injuries and fatalities can be caused by human, vehicle, road or environmental factors and can also be aggravated by several common causes, such as careless driving, dangerous overtaking, speeding, faulty vehicles, micro-sleeping while driving and poorly maintained roads [9,10]. With a daily average of 18 persons killed in road accidents in Malaysia, it may continue to be a serious public health challenge to the nation if no proactive actions are taken [8,11]. To formulate effective interventions to prevent and reduce the occurrence of non-fatal injuries (NFIs) due to RTAs, it is essential to proactively identify the factors that lead to RTAs. Therefore, this study aims to determine the prevalence of NFIs related to RTAs and factors associated with such injuries among adults in Malaysia.

## 2. Methodology

This was a sub-study under the Prospective Urban Rural Epidemiology (PURE) study that involved 21 countries, including Malaysia, and aimed to determine the impact of societal influences on the prevalence of selected non-communicable diseases. The comprehensive methodology of the overall PURE study has been explained in detail in published articles [12,13].

### 2.1. Study Population

This community-based study involved adults aged 35–70 years. Respondents were conveniently recruited from selected urban and rural areas. Urban and rural areas were randomly selected throughout peninsular and east Malaysia according to the definition by the Department of Statistics Malaysia (DOSM) [14]. Once the permission from the community leaders was obtained, health screening and promotion booths were set up in the communities’ assembly halls, where residents visited the booths. Interested and eligible participants were briefed about the study and upon obtaining verbal and written consent, medical histories were taken and basic physical examinations were conducted on the participants. All data were obtained through face-to-face interview sessions by well-trained research assistants using a standardised and verified set of questionnaires. Initially, 15,378 respondents agreed to participate in the study; however, only 15,321 (99.6%) individuals provided completed answered questionnaires (57 respondents were excluded).

### 2.2. Measures

The questionnaire was developed by the Population Health Research Institute (PHRI) team in Canada and was later thoroughly revised by the Malaysian team of researchers to ensure its suitability for the local settings. The basic questionnaire consisted of 30 questions covering socioeconomic characteristics, lifestyles and medication intake. Questions assessing depression were adopted from the short form of the Diagnostic and Statistical Manual of Mental Disorders (DSM)-IV Composite International Diagnostic Interview (CIDI) questionnaire. For body mass index (BMI), overweight was defined as greater or equal to 25 kg/m^2^ and less than 30 kg/m^2^ for obesity. Out of 15,378 respondents, 10,031 (65.2%) individuals gave consent for their blood pressure to be measured. NFIs were assessed by asking whether the participants had sustained any injuries during the previous 12 months that were serious enough to limit normal activities. Those who reported such injuries were asked to select the causes of the injuries from a list of 19 options, which included RTA (vehicle occupant/rider or pedestrian). They also provided data on urban/rural residence, age, sex, marital status, education, smoking habit, alcohol use, perceived stress, financial stress, depression and whether they practiced a noon nap (siesta).

### 2.3. Statistical Analysis

The data were analysed using the SPSS version 26 (IBM, Armonk, NY, USA). The general characteristics of respondents were descriptively analysed and presented as the numbers (and corresponding percentages). Multinomial logistic regression analysis was performed to investigate the potential determinants of RTAs. Odds ratios (ORs) and 95% confidence intervals (CIs) were calculated. Statistical significance was set at *p* < 0.05. The model was adjusted for age, gender, location, education level, socioeconomic status, marital and employment status, BMI, lifestyles, stress and depression.

### 2.4. Ethical Approval

The Hamilton Health Sciences Research Ethics Board has approved the study protocol (PHRI; grant no. 101414), with local ethical clearance from the Research and Ethics Committee Universiti Kebangsaan Malaysia (UKM) Medical Centre (project code: PHUM-2012–01) and the Research Ethics Committee of Universiti Teknologi Mara (UiTM).

## 3. Results

This study involved 15,321 respondents, with 863 (5.6%) reporting that they experienced at least 1 NFI in the past 12 months. Figure 1 shows the distribution of the reported NFIs, of which 35.1% were caused by RTAs, 30.1% were caused by falls and the remaining 34.8% were attributed to 15 other causes of NFIs that were too heterogeneous to be combined into larger meaningful groups.

Table 1 shows the general characteristics of the respondents who reported having an NFI due to an RTA. They were predominantly male (44.8%), had primary (35.8%) or secondary (39%) education, were currently employed (41.3%), were smokers (41.6%), were overweight to obese (36.7%), had perceived stress (40.1%), were on prescription medication (38%) and practiced a noon nap/siesta (37%).

The factors associated with NFI caused by RTA are shown in Table 2. Male respondents had twice the odds of experiencing injuries caused by RTA compared to female respondents (AOR 2.08; 95% CI 1.33–3.26). Those who reported having a primary (AOR 2.52; 95% CI 1.40–4.55) or secondary (AOR 2.64; 95% CI 1.55–4.49) education level were nearly three times more likely to experience injuries following RTAs, compared to those with a tertiary education level. Compared to those with a normal BMI, respondents who were overweight to obese were 1.4 times more likely to sustain an injury due to an RTA (AOR 1.40; 95% CI 1.01–1.94). In addition, respondents who were currently employed (AOR 2.03; 95% CI 1.31–3.13) had twice the odds of experiencing injuries caused by an RTA, compared to those who were currently unemployed. Regarding siestas, those who reported not practicing a noon nap were 1.4 times more likely to sustain injuries due to an RTA (AOR 1.38; 95% CI 1.01–1.89). However, factors such as age, geographical location, socioeconomic status, marital status, smoking, alcohol consumption, stress, depression, and medication intake showed no association with NFI caused by RTA.

## 4. Discussion

The prevalence of NFIs due to RTAs among this study population was 5.6%. According to the Malaysian Institute of Road Safety Research (MIROS), there was an increase in the number of RTAs involving injuries from 15,044 cases registered in 2019 to 17,236 cases in 2020, an increase of 14.6% [15]. From that total, approximately 57% (*n* = 9752) of the RTAs reported in 2020 involved light injuries, which is a huge spike from 5855 cases in 2019. Since the definition of RTA according to the Malaysia Road Accident Statistics Report from 2012 includes both injuries and fatalities due to vehicle collisions, data on fatalities are more pronouncedly reported as compared to injuries [16]. Over the last decade, Malaysia has recorded 4.94 million accidents, with the number of road accidents increasing from 414,421 cases in 2010 to 567,516 in 2019 [17,18]. Amid the COVID-19 pandemic, the country recorded 3118 deaths involving motorcycle riders and 888 deaths involving car drivers.

In this study, men exhibited a higher risk of experiencing an NFI due to an RTA than women. This result is supported by many local and international studies, in which a proposed explanation for this was the hormonal differences between genders [19,20,21,22,23]. Testosterone was suggested to influence the temperament of a driver by boosting aggression levels. This may not necessarily make men violent drivers, but it can induce greater risk-taking on the road and the tendency to feel overconfident. In addition, studies have shown that male drivers spent a greater amount of time behind the wheel, used safety gear less frequently and had an increased likelihood of speeding and/or driving while intoxicated [24,25,26,27]. Moreover, the use of two-wheeled vehicles in Malaysia is more common among men than women, as the latter tend to prefer either cars or public transportation [28,29]. Motorcycle riders have a much higher risk of injuries when involved in RTAs as compared to other vehicles, due to increased exposure and likelihood of direct physical contact during accidents. This is in line with the recent national RTA statistics, which showed that casualties from motorcycle accidents constitute 66% (31,222) of all traffic accident casualties (47,012) [8].

This study also found that individuals with a lower education level were three times more likely to sustain NFIs due to RTAs, compared to those with a higher education level. In addition to the reports suggesting that those with a lower education level are less likely to obey traffic laws and use seat belts, the majority of the literature has linked lower education levels with lower economic and cultural status, which makes it more difficult to own more expensive vehicles with higher safety standards [30,31,32]. Moreover, due to their economic constraints, they may hold multiple jobs that require working for a longer period of time without adequate rest. This may lead to sleep deprivation, which has been shown to be the leading contributor to RTAs and fatalities related to human factors [33,34]. Some studies have also reported that very few of those with a lower education level have any sort of health insurance [35,36,37]. Thus, any injuries, especially NFIs, that could have been caused by an RTA were seldom treated properly, which could result in worsening of the injuries if being involved in another RTAs. 

Compared to those with a normal BMI, the respondents who were overweight to obese were more likely to sustain NFIs due to RTAs. Although studies have successfully linked higher BMI with injuries due to RTAs [38,39], a U-shaped relationship has been established between these two variables [40,41,42]. However, it has been highlighted that overweight to obese individuals are more prone to momentum effects when involved in an RTA, as their weight is more likely to propel them with greater force into the steering wheel, dashboard or car window, leading to injuries. They may also experience higher impact during the RTA, increasing the risk of injuries as well as fatalities. Moreover, individuals with a high BMI commonly have co-morbidities, such as hypertension or diabetes, which may induce fatigue due to sleep apnoea or impaired driving and braking responses due to nerve damage to the feet, which is associated with having diabetes [43]. In addition, seat belts were designed for the average driver, and thus may offer a poor fit for those who are overweight or obese [44,45]. Thus, seat belts may be less effective in protecting these individuals when deployed, and obese drivers may be less likely to wear them due to discomfort. These factors increase the risk of injuries when an RTA occurs.

Employed individuals in this study were shown to be more susceptible to injuries due to RTAs than those who were unemployed. This may be due to the higher frequency of using vehicles and being on the road among employed individuals. Indeed, there are even roles that require the employee to be behind the wheel for the entire time on the job, such as long-distance lorry or bus drivers, taxi drivers, food or parcel delivery personnel and postal workers. These types of work, especially making food or parcel deliveries, usually demand speed and lead to drivers being involved in traffic violations and recklessly driving in order to fulfil as many delivery requests as possible to obtain a higher income [46,47]. On the other hand, other employed individuals also use vehicles to commute to and from work via motorcycles, cars or public transportation. In urban areas, motorcycles are the chosen mode of transportation, as they allow the driver to avoid heavy traffic. Studies have shown that more than 50% of road accidents in Malaysia involved motorcyclists and that their risk of suffering injuries is higher compared to occupants of other vehicles [48,49].

A siesta or noon nap was shown to be a protective factor against NFIs due to RTAs. This was supported by studies that found that road users who reported practicing noon naps were less prone to feeling fatigue, and hence had a lower propensity to engage in risky driving behaviours, such as speeding, that can lead to injuries [36,50,51]. Napping for at least 20 min combined with caffeinated drinks was shown to be the most effective method (versus caffeine alone) to reduce sleepiness among both short- and long-distance drivers [52]. This strategy was reported to significantly improve drivers’ accuracy, reaction speed and cognitive functioning [53,54,55]. A comprehensive review on studies related to afternoon napping concluded that the so-called ‘post-lunch dip’ is a period of sleepiness that occurs between 13:00 and 16:00 following the human circadian rhythm, resulting in peak performance occurring in the morning and early evening and lower performance in the afternoon [50]. This is also the period of time in which there is a slight reduction in core body temperature, which is said to promote a tendency to sleep and cause a temporary decrease in vigilance, thus leading to drowsiness, and may cause RTAs [56,57]. Thus, having a nap during the most unproductive afternoon period has been shown to boost alertness later in the evening, reducing the risk of micro-sleeping while driving, which may lead to RTAs, and thereby cause preventable injuries.

Apart from the traditional approach on reducing the occurrence of RTA by focusing on the behavioural changes of the road users, the Safe System (SS) approach has also been implemented and has been growing ever since the Malaysia government launched the Global Plan for the Decade of Action for Road Safety 2021–2030 that targeted a 50% reduction in road death and serious injuries by 2030 [58]. Infrastructural changes, such as central hatching on the road instead of a double line, replacing a proposed signalized intersection with a roundabout and more horizontal deflection upon approaching pedestrian crossing points, are some of the adjustments made using the SS approach [11,59]. Apart from that, the local authorities also have an active schedule of roadside treatment to remove hazards, such as big and rotten tree trunks, low hanging branches and overgrown vegetation covering traffic signs and signals and prompt potholes patching once reported or detected by traffic cameras.

Although there have been a few studies showing that the occurrence of NFIs due to RTA is higher in urban compared to rural area, this study did not show such a result [60,61]. This may be explained by better road accessibility in rural areas compared to the last decade due to aggressive land opening for agricultural purposes [62,63,64]. Thus, this may result in the equal risk of NFIs due to RTA in both urban and rural areas. Furthermore, the possibility of receiving rapid and advanced treatment once a RTA has happened is much faster in urban areas as opposed to rural areas, which may result in a lower probability of injuries causing fatalities. 

There are several limitations to our study. First, this study was a cross-sectional study in which the temporal link between the outcome and the exposure cannot be determined because both were examined at the same time. Second, the causes of NFI were reported by the respondents and may be subject to recall bias. However, during the interview with the respondents, the researchers clearly explained that the study is restricted to only NFI, which to an extent limited their responses. Apart from this, this study managed to involve more than 15,000 participants covering the entirety of Malaysia.

## 5. Conclusions

This study highlighted that NFIs due to RTAs were associated with being male, having a primary or secondary education level, being overweight to obese, being employed and not taking a noon nap. The occurrence of NFIs due to RTAs is highly preventable with strategic planning by policymakers aimed at reducing the risk of RTAs among the Malaysian population. Simple interventions focused on protecting road users, especially those who drive two-wheelers, with proactive road safety awareness and literacy campaigns, combined with strict enforcement of the existing traffic laws and behavioural modifications, may lower the risk of NFIs following RTAs at the national level.

## Figures and Tables

**Figure 1 ijerph-19-08246-f001:**
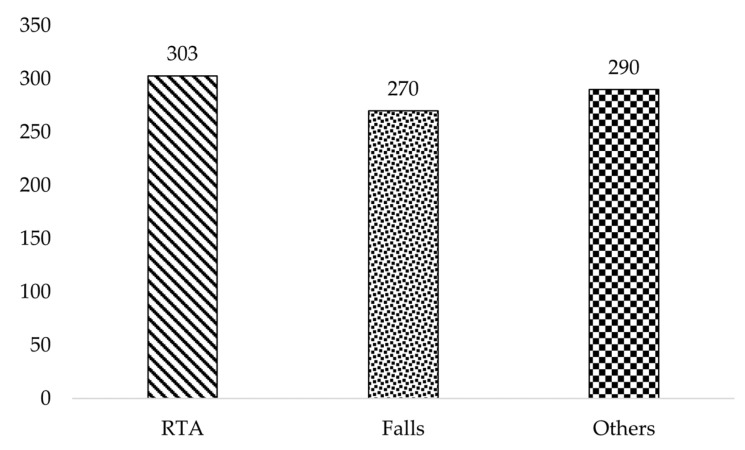
Distribution of causes of non-fatal injuries (*n* = 863).

**Table 1 ijerph-19-08246-t001:** General characteristics of study participants (*n* = 863).

Risk Factors		Road Traffic Accident		*p*-Value
		Yes *n* (%)	No *n* (%)	
		303 (35.1)	560 (64.9)	
Age (years old)	35–40	40 (32.0)	85 (68.0)	0.644
	41–50	90 (33.7)	177 (66.3)	
	51–60	102 (35.7)	184 (64.3)	
	61–70	71 (38.4)	114 (61.6)	
Gender	Female	104 (24.8)	315 (75.2)	<0.001 **
	Male	199 (44.8)	245 (55.2)	
Education level	Primary	141 (35.8)	253 (64.2)	0.002 *
	Secondary	137 (39.0)	214 (61.0)	
	Tertiary	25 (21.2)	93 (78.8)	
Socioeconomic status	Low	133 (35.8)	238 (64.2)	0.330
	Middle	153 (35.7)	275 (64.3)	
	High	17 (26.6)	47 (73.4)	
Marital status	Currently unmarried	25 (28.1)	64 (71.9)	0.138
	Currently married	278 (36.0)	494 (64.0)	
Employment status	Yes	238 (41.3)	338 (58.7)	<0.001 **
	No	65 (22.6)	222 (77.4)	
Location	Urban	104 (31.4)	227 (68.6)	0.079
	Rural	199 (37.4)	333 (62.6)	
BMI	Normal	95 (32.1)	201 (67.9)	0.202
	Overweight (obese)	208 (36.7)	359 (63.3)	
Smoking status	Yes	179 (31.7)	386 (68.3)	0.002 *
	No	124 (41.6)	174 (58.4)	
Alcohol consumer	Yes	283 (35.2)	520 (64.8)	0.889
	No	20 (33.3)	40 (66.7)	
Perceived stress	Yes	119 (40.1)	178 (59.9)	0.030 *
	No	183 (32.6)	379 (67.4)	
Financial stress	Yes	56 (37.1)	95 (62.9)	0.575
	No	247 (34.7)	465 (65.3)	
Depression	Yes	280 (35.9)	499 (64.1)	0.128
	No	20 (26.7)	55 (73.3)	
On medication	Yes	184 (38.0)	300 (62.0)	0.044 *
	No	117 (31.3)	257 (68.7)	
Siestas	Yes	190 (37.0)	324 (63.0)	0.168
	No	113 (32.4)	236 (67.6)	

* Significant at *p*-value < 0.05; ** Significant at *p*-value < 0.001

**Table 2 ijerph-19-08246-t002:** Factors associated with road traffic accidents (*n* = 863).

Variables		B	S.E	AOR (95% CI)	*p*-Value
Age (years old)	35–40			1	
	41–50	−0.061	0.249	0.941 (0.577–1.533)	0.806
	51–60	0.078	0.264	1.081 (0.645–1.814)	0.767
	61–70	0.175	0.295	1.191 (0.668–2.122)	0.554
Gender	Female			1	
	Male	0.732	0.23	2.079 (1.325–3.263)	<0.001 **
Location	Rural	0.346	0.229	1.413 (0.902–2.214)	0.132
	Urban			1	
Education level	Primary	0.925	0.301	2.522 (1.398–4.549)	0.002 *
	Secondary	0.97	0.271	2.637 (1.551–4.486)	<0.001 **
	Tertiary			1	
Socioeconomic status	Low	−0.209	0.398	0.811 (0.372–1.768)	0.598
	Middle	0.073	0.337	1.076 (0.556–2.08)	0.828
	High			1	
Marital status	Currently unmarried	0.032	0.271	1.033 (0.608–1.755)	0.906
	Currently married			1	
BMI	Normal			1	
	Overweight (obese)	0.333	0.167	1.396 (1.006–1.937)	0.046 *
Smoking status	Yes	−0.234	0.196	0.791 (0.539–1.162)	0.233
	No				
Alcohol consumption	Yes	−0.129	0.316	0.879 (0.473–1.633)	0.683
	No				
Employment status	Yes	0.707	0.223	2.028 (1.310–3.140)	0.002 *
	No				
Perceived stress	Yes	−0.26	0.17	0.771 (0.552–1.076)	0.126
	No			1	
Financial stress	Yes	−0.03	0.211	0.971 (0.641–1.469)	0.888
	No			1	
Clinical depression	Yes	−0.127	0.291	0.88 (0.498–1.556)	0.661
	No			1	
On medication	Yes	−0.289	0.162	0.749 (0.545–1.029)	0.075
	No			1	
Siesta (noon nap)	Yes			1	
	No	0.321	0.16	1.378 (1.007–1.887)	0.045 *
Constant		−2.535	0.516		<0.001 *

* Significant at *p*-value < 0.05; ** Significant at *p*-value <0.001; R^2^ = 12.8%.

## Data Availability

The data presented in this study are available on request from the corresponding author.

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
