# Peer review of "Assessing Factors Associated with Non-Fatal Injuries from Road Traffic Accidents among Malaysian Adults: A Cross-Sectional Analysis of the PURE Malaysia Study"

_ijerph, 2022, doi:10.3390/ijerph19148246_

Round 1
Reviewer 1 Report
Road traffic injuries and deaths in Malaysia are a growing problem. Most prevention programmes are based on death data as representative non-fatal injury data is costly and requires large sample sizes. To circumvent these challenges the authors have analyzed data from the PURE project to ascertain the incidence of non-fatal injuries sustained in the last 12 months and associated risk factors. The methods and analysis that have been conducted are appropriate and show an increased odds of sustaining an NFI related to being male, lower level of education, working population (all to be expected) and unexpectedly not taking a nap in the afternoon. The conclusions drawn from the analysis point to the need to addressing these behavioural factors and more stringent enforcement of current road safety laws. While these interventions are required, it would have been good to see how they fit into a broader Safe Systems Approach to road safety which focuses less on individual behaviour and more on how the system can accommodate human error. It would have been useful to see some discussion on this paradigm which doesnt blame the victim. Some discussion about potential infrastructural changes or vehicle technologies to reduce road traffic crashes could be introduced to balance the debate and show that Malaysia is moving away from the old behavioural only approach.
Author Response
Dear reviewer,
Attached is the point-by-point response.
Thank you for your comments.

Reviewer 2 Report
Kindly see the attached.

Author Response
Dear reviewer,
Attached is the point-by-point response.
Thank you for the comments.

Round 2
Reviewer 2 Report
The authors well addressed comments from the reviewer. Good job.